# NEXT TOKEN PERCEPTION SCORE: ANALYTICAL ASSESSMENT OF YOUR LLM PERCEPTION SKILLS

## ABSTRACT

Autoregressive pretraining has become the de facto paradigm for learning general-purpose representations in large language models (LLMs). However, linear probe performance across downstream perception tasks (e.g., classification, regression) shows substantial variability, suggesting that features optimized for next-token prediction do not consistently transfer well to downstream perception tasks. We demonstrate that representations learned via autoregression capture features that may lie outside the subspaces most informative for perception. To quantify the (mis)alignment between autoregressive pretraining and downstream perception, we introduce the Next Token Perception Score (NTPS), a score derived under a linear setting that measures the overlap between autoregressive and perception feature subspaces. This metric can be efficiently computed in closed form from pretrained representations and labeled data, and is proven to both upper- and lower-bound the excess loss. Empirically, we show that NTPS correlates strongly with linear probe accuracy across 12 diverse NLP datasets and eight pretrained models ranging from 270M to 8B parameters, confirming its utility as a measure of alignment. Additionally, NTPS reliably predicts the additional accuracy gains attained by LoRA finetuning thereby providing a lightweight prescreening tool for LoRA adaptation. Our results offer both theoretical insights and practical tools for analytically assessing LLM perception skills.

## 1 INTRODUCTION

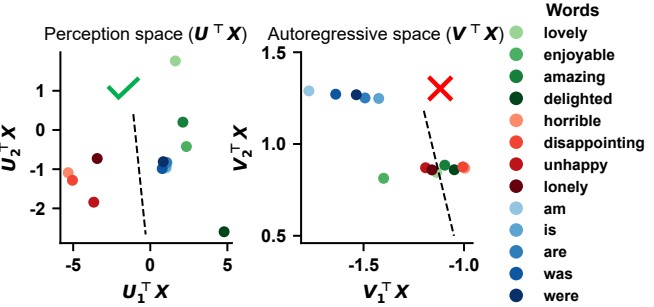

Figure 1: 2D projections of the representations learned through perception $U^\top X$ (left) or autoregressive $V^\top X$ (right) tasks based on OpenELM-450M and Emotion(Saravia et al., 2018) dataset. **In the perception space, joy and sad words are clearly separated, illustrating label-relevant semantic encoding; in the autoregressive space, these two emotion classes overlap, but syntactic categories become clearly clustered.**

The success of GPT-1 (Radford et al., 2018) demonstrated the effectiveness of autoregressive pretraining, training a model to predict the next token given preceding context, for learning transferable language representations. This autoregressive training paradigm quickly became the standard for building large language models (LLMs), leading to increasingly capable successors such as GPT-2/3/4 (Radford et al., 2019; Mann et al., 2020; Achiam et al., 2023) and LLaMA-1/2/3/4 (Touvron et al., 2023a;b; Grattafiori et al., 2024; Singh, 2025). As model capabilities scaled, the expectation evolved: rather than merely serving as a source of transferable features requiring task-specific finetuning, a foundation model is now expected to perform well on downstream perception tasks (e.g., classification, regression) straight out of the box, without modifying its pretrained weights.

Under this expectation, prompting has emerged as a popular strategy (Radford et al., 2019), adapting the model to downstream perception tasks solely by crafting input text. Meanwhile, linear probing has gained traction as an alternative (Kumar et al., 2022; Liu et al., 2023), leveraging frozen representations and training only a lightweight linear classifier on top, offering a more structured and efficient approach to downstream adaptation. While prompting and linear probing may appear quite different in practice, they are fundamentally two sides of the same coin. In prompting, natural language input steers the model, and the model's pretrained linear head for autoregression maps hidden states to output token probabilities. In linear probing, by contrast, the model is frozen and an external linear head is trained to interpret its hidden representations.

However, as we show in section 3, linear probing does not perform equally well across all downstream perception tasks. In some cases, training a model from scratch on the downstream dataset yields significantly better performance. This suggests that representations learned by current LLMs are not universally effective: while some downstream tasks benefit greatly from pretraining and are well-served by simple adaptation methods like linear probing, others are not. Motivated by this observation, we pose the following question:

*How can we quantify the alignment between autoregressive pretraining and downstream perception tasks to explain the varying effectiveness of linear probing across different datasets?*

In this paper, we take a first step towards understanding and assessing the alignment between autoregressive pretraining and downstream perception tasks by:

- **Systematically evaluating the benefits of autoregressive pretraining**—by comparing linear probe performance on six pretrained models against identical architectures trained from scratch across 12 downstream perception datasets.

- **Proposing the Next Token Perception Score (NTPS)**—a metric that quantifies the alignment between autoregressive pretraining and downstream perception tasks by measuring the overlap between their respective feature subspaces.

- **Empirically validating the reliability of NTPS**—by demonstrating that it correlates strongly with linear probe accuracy across 12 diverse datasets and eight pretrained models.

- **Guiding LoRA finetuning with NTPS**—by demonstrating that NTPS reliably forecasts the additional accuracy gains from LoRA finetuning, thereby providing a lightweight prescreening tool.

The remainder of the paper is organized as follows. Section 2 reviews both empirical and theoretical foundations for applying pretrained LLMs directly to downstream tasks. Section 3 presents evidence that pretrained LLM representations are not universally effective when used directly through linear probing and therefore we introduce our proposed NTPS metric to quantify such (mis)alignment. Section 4 empirically demonstrates that NTPS correlates with linear probing performance across 12 downstream datasets and eight pretrained models and NTPS can serve as a good predictor of performance gain from LoRA finetuning. Section 5 summarizes our findings.

## 2   BACKGROUND

**Utilizing Pretrained LLM Hidden Representations for Downstream Tasks**   Pretrained large language models can be leveraged for downstream tasks without any gradient-based finetuning via two complementary strategies: prompt-based in-context learning and linear probing of fixed hidden representations. In prompting, GPT-3 attains strong zero- and few-shot classification performance across diverse NLP benchmarks using only natural language templates and without any weight updates (Mann et al., 2020). Likewise, when given only in-context examples, GPT-3 can perform both linear and non-linear regression at levels comparable to Random Forests and Gradient Boosting (Vacareanu et al., 2024). In linear probing, early work reveals that deep linguistic structures are capturable by training simple classifiers on fixed frozen representations (Tenney et al., 2019; Jawahar et al., 2019). Recent work show that LLM embeddings preserve Lipschitz continuity and can be utilized in high-dimensional regression settings, outperforming conventional feature-engineering baselines (Tang et al., 2024).

**Scaling laws for Predicting Downstream Performance in LLMs**    The downstream performance of LLMs has garnered significant attention, with special focus on scaling laws. Gadre et al. (2024) have found a power law between validation loss and model FLOPs, and also a power law between language model perplexity and average performance across all datasets. Isik et al. (2025) have found a log law in translation tasks between downstream translation performance and the number of tokens in the pretrained task. Chen et al. (2024) have proposed a two-stage approach to predict downstream performance: first mapping computational resources to pretraining loss based on a power law, then mapping pretraining loss to downstream task performance based on a linear mapping. Although these formulas achieve reasonable forecasts, they still rely on finetuning smaller models for calibration and offer limited mechanistic insight into why certain tasks benefit more from scale.

**Metrics-Based Approaches for Predicting Transferability**    In parallel to scaling-law analyses, several works have proposed metrics that directly estimate the transferability of pretrained representations without finetuning. Examples include LEEP (Nguyen et al., 2020), H-score (Bao et al., 2019), and TransRate (Huang et al., 2022), which quantify information content in fixed hidden embeddings and correlate with downstream performance across benchmarks. While these approaches offer practical tools for model selection and evaluation, they remain descriptive rather than explanatory: they assess transfer efficacy by probing frozen representations, but do not account for how autoregressive pretraining shapes these representations in the first place.

**Theoretical Foundations of Utilizing Pretrained LLM Hidden Representations**    Recent studies in understanding pretraining objectives have revealed precise conditions under which different self-supervised losses guarantee, or fail to guarantee, strong downstream performance. Balestriero & LeCun (2024) rigorously demonstrate that reconstruction-based training, such as autoencoders or masked reconstruction, can produce features that capture all input variance yet remain uninformative for discriminative perception, underscoring that low reconstruction error alone is insufficient for transfer. Wu et al. (2023) identify two necessary conditions for autoregressive next-token models to transfer effectively: the softmax head must break shift invariance, and downstream tasks must not hinge on tokens with vanishingly small pretraining probabilities. Liu et al. (2023) show that among models with identical pretraining loss, those converging to flatter minima generalize best, revealing that the implicit bias of optimization plays a crucial role in shaping downstream performance.

## 3    NEXT TOKEN PERCEPTION SCORE (NTPS): AN ANALYTICAL ASSESSMENT METRIC OF PRETRAINED LLMS PERCEPTION SKILLS

In this section, we first present empirical evidence in section 3.1 showing that while pretrained LLM representations can enhance performance on certain downstream perception tasks, they may underperform or provide no advantage on others compared to models trained from scratch. We then demonstrate in a linear regime that such variability can be explained by the extent to which the perception feature subspace aligns with the autoregression feature subspace, and introduce the Next Token Perception Score (NTPS) to quantify this relationship in section 3.2. Finally, in section 3.3, we show that NTPS serves as a valid and efficient proxy for downstream performance in practice.

### 3.1    ON THE NEED TO MONITOR LLM ALIGNMENT FOR PERCEPTION TASK

To demonstrate that LLM representations are not universally effective, we compare the linear probe performance of pretrained models on downstream perception tasks with the performance of the same architectures trained from scratch on the downstream datasets.

Here, we evaluate six models: Qwen2-0.5B/1.5B (Yang et al., 2024) and OpenELM-270M/450M/1.1B/3B (Mehta et al., 2024). The evaluation spans 12 downstream datasets across a variety of domains, including: Intent Classification (Bhuvaneshwari, 2022), Clickbait Title Classification (Chakraborty et al., 2016), SST-2 (Socher et al., 2013), Bias Identification (Patel, 2023), Banking(Casanueva et al., 2020), Emotion(Saravia et al., 2018), SMS Spam (Almeida et al., 2011), Medical Question Pairs (McCreery et al., 2020), Rotten Tomatoes (Pang & Lee, 2005), CommonsenseQA (Talmor et al., 2019), Climate Sentiment (Bingler et al., 2023), and IMDB (Maas et al., 2011).

Table 1: Comparison of linear probe performance of pretrained models versus full-training from scratch across downstream datasets. **Linear probing can outperform, match, or underperform full-training from scratch, indicating that pretrained LLM representations are not universally effective.**

| | Qwen2 0.5B | | Qwen2 1.5B | | OpenELM 270M | | OpenELM 450M | | OpenELM 1.1B | | OpenELM 3B | |
|---|---|---|---|---|---|---|---|---|---|---|---|---|
| | Linear | Full | Linear | Full | Linear | Full | Linear | Full | Linear | Full | Linear | Full |
| Intent | 99.7 | 99.6 | 99.9 | 99.5 | 99.3 | **99.6** | 99.5 | **99.6** | 99.8 | 99.8 | 98.4 | **99.0** |
| Clickbait Title | 99.4 | 99.1 | 99.6 | 99.0 | 99.4 | 98.4 | 99.6 | 98.4 | 99.7 | 98.7 | 99.6 | 98.6 |
| SST-2 | 85.4 | 80.4 | 88.2 | 82.1 | 87.6 | 80.3 | 87.7 | 82.5 | 89.3 | **92.0** | 89.9 | 78.7 |
| Banking | 88.1 | 85.4 | 89.4 | 82.4 | 89.8 | 86.3 | 90.5 | 84.8 | 91.3 | 83.3 | 82.0 | **82.3** |
| Bias | 95.5 | 94.9 | 96.4 | 94.6 | 96.5 | 94.7 | 96.4 | 94.4 | 96.8 | 95.5 | 95.4 | 94.8 |
| Emotion | 66.2 | **88.3** | 69.0 | **88.3** | 70.9 | **86.8** | 72.0 | **86.7** | 73.6 | **76.9** | 63.6 | **87.9** |
| SMS Spam | 99.3 | 98.7 | 99.3 | 98.9 | 99.0 | 98.8 | 99.2 | 98.7 | 99.2 | 98.9 | 98.4 | **99.0** |
| Medical | 36.4 | **51.5** | 28.9 | **51.3** | 33.8 | **51.5** | 30.6 | **51.5** | 27.5 | **51.5** | 36.7 | **51.5** |
| Rotten Tomatoes | 81.9 | 75.9 | 85.6 | 73.5 | 82.6 | 74.1 | 84.1 | 76.9 | 86.8 | 75.2 | 84.8 | 74.9 |
| Commonsense | 22.1 | 21.0 | 24.2 | 22.4 | 22.3 | 21.2 | 21.2 | **22.2** | 23.3 | 21.3 | 21.5 | 21.3 |
| Climate | 78.4 | 63.7 | 81.3 | 67.8 | 80.3 | 69.1 | 79.1 | 71.9 | 81.6 | 69.1 | 79.4 | 71.2 |
| IMDB | 92.5 | 86.0 | 94.4 | 84.9 | 92.8 | 84.2 | 99.5 | 84.1 | 94.5 | 83.5 | 94.4 | – |

For full training, we use the same configuration as in Balestriero & Huang (2024) across all models and datasets: Adafactor optimizer with a learning rate of $10^{-4}$, $\epsilon$ values of $10^{-30}$ and $10^{-3}$, gradient clipping threshold of 1.0, decay rate of 0.8, and weight decay of $10^{-5}$; $10,000$ training steps with a cosine learning rate scheduler with a 5% warm-up phase. For linear probing: we use the following configuration across all models and datasets: AdamW optimizer with a learning rate of $10^{-4}$; 50 epochs. For both cases, we extract the mean token representation from the final transformer block and feed it into a linear classification head. Training losses for full training can be seen in fig. S1.

As shown in table 1[1], the effect of autoregressive pretraining with linear probing varies markedly across datasets. On sentiment-analysis tasks including SST-2 (Socher et al., 2013), Rotten Tomatoes (Pang & Lee, 2005), Climate (Bingler et al., 2023) and IMDB (Maas et al., 2011), linear probing delivers gains of roughly 5–10%. For intent classification (Bhuvaneshwari, 2022), clickbait detection (Chakraborty et al., 2016), bias identification (Patel, 2023), SMS spam (Almeida et al., 2011) and CommonsenseQA (Talmor et al., 2019), the performance difference between linear probing and training from scratch stays within about 1%. In the most extreme cases, emotion recognition (Saravia et al., 2018) and medical-text classification (McCreery et al., 2020), linear probing actually underperforms training from scratch by a substantial margin.

This variability suggests that the representations learned via autoregressive pretraining do not uniformly align with downstream perception tasks. Therefore, we are going to quantify this alignment (or lack thereof). As a starting point, we first build intuition and establish theoretical results in the *linear regime*. As we will show in section 4, this seemingly simplified setting provides surprisingly informative insights into more complex empirical scenarios.

> **Takeaway**: Linear probing on pretrained LLM representations can outperform, match, or underperform full-training from scratch.

### 3.2 QUANTIFYING ALIGNMENT IN THE LINEAR REGIME WITH NTPS

Consider a sentence, whose representation is $X \in \mathbb{R}^{(d,\ell)}$, where $d$ is the hidden dimension size of each token and $\ell$ is the total number of tokens in this sentence. Consider two variants $X_1, X_2 \in \mathbb{R}^{d \times (\ell-1)}$ from $X$, where $i$-th column of $X_1$ is the representation of the first $i$ tokens of the sentence and $i$-th column of $X_2$ is the representation of the $i + 1$-th token of the sentence.

---

[1]The full training record for OpenELM 3B is unavailable due to insufficient GPU memory (A100), even when using a batch size of 1.

Autoregressive training aims to find a model's parameter $\theta$ to predict $X_2$ based on $X_1$. Specifically, the training objective is to minimize the following Cross-Entropy(CE) loss:

$$\mathcal{L}_{\text{CE}} = -\mathbb{E}_X\big[\log p_\theta(X_2 \mid X_1)\big], \qquad p_\theta(X_2 \mid X_1) \propto g_\theta(f_\theta(X_1)). \tag{1}$$

In the linear setting, $f_\theta$ is a linear map $V \in \mathbb{R}^{d \times k}$ and $g_\theta$ is another linear map $W \in \mathbb{R}^{k \times d}$. Besides, we assume the $i$-th column of $X_1$ represents the sum of the first $i$ tokens in $X$ and the $i$-th column of $X_2$ represents the $i+1$-th token in $X$.

Then the loss function $\mathcal{L}$ is defined as:

$$\mathcal{L} = \mathbb{E}_{X_1,X_2}\big\|W^\top V^\top X_1 - X_2\big\|_F^2 = \mathbb{E}_X\big\|W^\top V^\top X L_1 - X L_2\big\|_F^2. \tag{2}$$

Here $L_1, L_2 \in \mathbb{R}^{\ell \times (\ell-1)}$ is for selecting the tokens in $X$, see section A.2 for full definition.

Similarly, given another pair $(U \in \mathbb{R}^{d \times k}, Z \in \mathbb{R}^{k \times c})$. When we use the sum of all tokens in the sentence to predict the label $Y \in \mathbb{R}^c$, our loss $\mathcal{L}^*$ is defined below:

$$\mathcal{L}^* = \mathbb{E}_{X,Y}\big\|Z^\top U^\top X \, 1_{\ell \times 1} - Y\big\|_F^2, \tag{3}$$

Note that in both cases, instead of the CE loss, we assume a mean square error (MSE) loss. We now state a key guarantee for our choice: as the MSE loss vanishes, so does the probability of decoding error.

**Lemma 1** (Equivalence between MSE and CE; proof in section A.1, empirical validation in (Hui & Belkin, 2020)). *Let $X \in \mathbb{R}^{d \times \ell}$ be the token representations. Denote*

$$h^* = X L_2, \qquad \hat{h} = W^\top V^\top X L_1.$$

*Assume the vocabulary embeddings $\{w_i\}_{i=1}^V \subset \mathbb{R}^d$ satisfy a positive margin*

$$\Delta = \min_{j \neq y}\langle w_y, h^*\rangle - \langle w_j, h^*\rangle > 0, \tag{4}$$

*If $\mathcal{L} = \mathbb{E}\|\hat{h} - h^*\|_F^2 \to 0$, then*

$$\Pr\big(\arg\max_i \langle w_i, \hat{h}\rangle = y\big) \longrightarrow 1. \tag{5}$$

Now we can solve the eq. (2) and eq. (3) under the following theorem.

**Theorem 1** (proof in section A.2). *The loss functions $\mathcal{L}$ in eq. (2) and $\mathcal{L}^*$ in eq. (3) are minimized for*

$$W = (V^\top \mathbb{E}[X L_1 L_1^\top X]V)^{-1}V^\top \mathbb{E}[X L_1 L_2^\top X] \tag{6}$$

$$Z = (U^\top \mathbb{E}[X \, 1_{\ell \times 1}1_{\ell \times 1}^\top X]U)^{-1}U^\top \mathbb{E}[X \, 1_{\ell \times 1}Y^\top] \tag{7}$$

*$U, V$ span the top $k$ eigenvectors of the following generalized eigenvalue problems:*

$$\mathbb{E}[X L_1 L_2^\top X^\top]\,\mathbb{E}[X L_2 L_1^\top X^\top]\,\tilde{V} = \mathbb{E}[X L_1 L_1^\top X^\top]\,\tilde{V}\,\Lambda_V, \tag{8}$$

$$\mathbb{E}[X \, 1_{\ell \times 1} Y^\top]\,\mathbb{E}[Y \, 1_{1 \times \ell} X^\top]\,\tilde{U} = \mathbb{E}[X \, 1_{\ell \times \ell} X^\top]\,\tilde{U}\,\Lambda_U. \tag{9}$$

From theorem 1, it shows that $U$ and $V$ capture distinct co-variability structures; hence, the autoregressively derived $V$ may not generalize well to downstream tasks that depend on $U$. To illustrate this, we extract the first embedding layer activations from a pretrained models (OpenELM-450M) on the Emotion(Saravia et al., 2018) dataset. We then solve the two generalized eigenvalue problems in theorem 1 to obtain the projection matrices $U$ and $V$, each truncated to its top two eigenvectors.

Figure 1 visualizes two-dimensional projections of representative words under the perception ($U^\top X$) and autoregressive ($V^\top X$) mappings. In the left panel ($U$-space), points colored by emotion form two nearly linearly separable clusters, positive versus negative, reflecting the label-conditioned objective. The right panel shows that the same words in $V$-space overlap heavily across emotion classes, indicating that next-token prediction does not prioritize emotional polarity. Instead, the $V$-space shows a clear grouping of adjectives vs. function words, suggesting that autoregressive training emphasizes syntactic category. In short, $U$ specializes in downstream, label-relevant semantics such as sentiment, whereas $V$ encodes the syntactic information essential for language modeling.

To obtain a continuous measure of overlap between the feature subspaces learned by $U$ (perception) and $V$ (autoregressive), we introduce the following alignment score. Let $P = V V^\dagger$ be the orthogonal projector onto the column space of $V$, where $V^\dagger$ is the Moore–Penrose pseudoinverse of $V$. We then define

$$\mathrm{NTPS}(U, V) = \frac{\|PU\|_F^2}{\|U\|_F^2}.$$

Here, $\|PU\|_F^2$ is the total squared projection of $U$ onto $V$'s subspace, and $\|U\|_F^2$ is the total variance in $U$. By construction, $0 \leq \mathrm{NTPS}(U, V) \leq 1$, achieving 1 if and only if the column spaces of $U$ and $V$ coincide, and approaching 0 as they become orthogonal. Higher values thus indicate greater alignment between the two objectives. The pseudocode for NTPS is provided in algorithm 1.

**Theorem 2** (Excess regression loss bounded by NTPS; proof in section A.3). *Let $U$ be the optimal perceptual encoder obtained from the generalized eigenproblem. For any other encoder $V$, define the orthogonal projector. Let $\Delta\mathcal{L} := \mathcal{L}^*(V) - \mathcal{L}^*(U)$ be the excess regression loss of $V$.*

*Then there exist task-dependent positive constants $C_{\min}, C_{\max}$ such that*

$$C_{\min}\big(1 - \mathrm{NTPS}(U, V)\big) \leq \Delta\mathcal{L} \leq C_{\max}\big(1 - \mathrm{NTPS}(U, V)\big). \tag{10}$$

This shows that the extra regression loss of the encoder $V$ trained with autoregression over $U$ trained with perception is tightly controlled by their subspace alignment: as $\mathrm{NTPS}(U, V)$ approaches 1 (perfect alignment), the excess loss vanishes, and as NTPS decreases, the loss grows linearly within the constant bounds.

> **Takeaway**: Our NTPS alignment score quantitatively captures how much of the perception-trained subspace lies in the autoregressive subspace and is proved to bound the excess loss.

### 3.3 NTPS AS A VALID AND EFFICIENT PROXY FOR DOWNSTREAM PERFORMANCE

Although NTPS is derived in the linear regime, the sentence representation $X$, when taken from intermediate layers of a nonlinear model, already encodes rich nonlinearities. As a result, applying NTPS to such representations preserves the theoretical validity established in the linear setting, while also benefiting from the expressive power of nonlinear architectures in practice.

Moreover, the calculation of NTPS is highly efficient since no learning or backpropagation is involved. Computing the expectation terms in the generalized eigenvalue problem of theorem 1 involves only a forward pass, with time complexity $\mathcal{O}(n)$ (where $n$ is the dataset size) and memory complexity $\mathcal{O}(b)$ (where $b$ is the batch size). Solving the generalized eigenvalue problem itself costs $\mathcal{O}(d^3)$ time and $\mathcal{O}(d^2)$ memory, where $d$ is the hidden dimension, both independent of dataset size and training epochs. In contrast, directly assessing downstream performance, either via linear probing or after finetuning with LoRA, scales with the number of training epochs. LoRA further requires backpropagation through the full network, typically incurring more than twice the computational cost of a forward pass (Wiedemann et al., 2020). While linear probing can reduce compute by caching frozen features, this comes at the cost of $\mathcal{O}(n)$ memory, with $n \gg b$ in most practical scenarios.

> **Takeaway**: NTPS offers a valid and efficient alternative to costly probing or finetuning for estimating downstream performance.

### 4 EMPIRICAL VALIDATION AND PRACTICAL UTILITY OF NTPS

In this section, we show that our NTPS, though derived under a linear model, captures meaningful alignment in nonlinear large-scale LLMs. First, section 4.1 demonstrates a monotonic relationship between NTPS and downstream performance by showing Spearman correlations with both MSE loss and classification accuracy across eight pretrained models and 12 downstream perception datasets. Next and more importantly, in section 4.2, we demonstrate that NTPS itself can predict the magnitude of accuracy gains from LoRA, making it a practical pre-screening metric for when finetuning will be most effective.

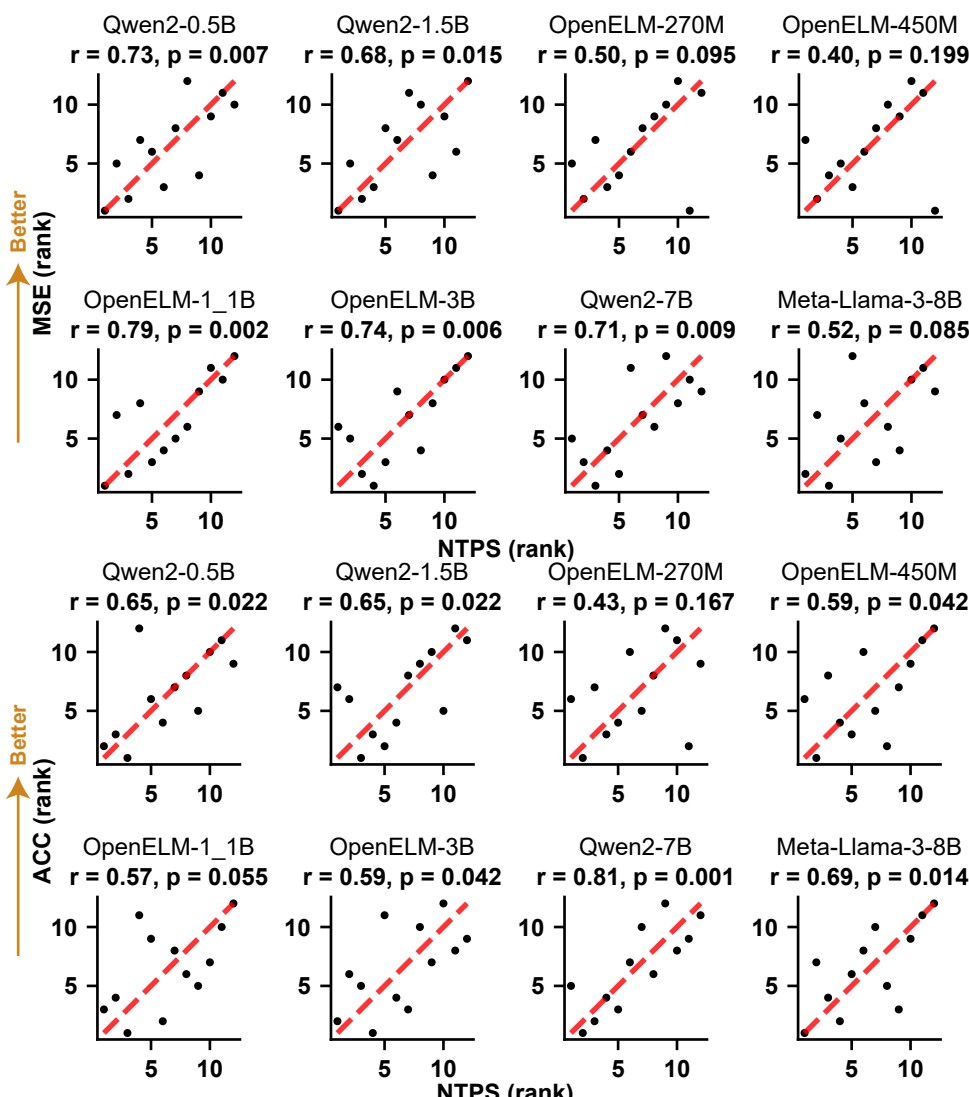

Figure 2: Correlation between NTPS and downstream MSE loss (rows 1 and 2), and between NTPS and accuracy (rows 3 and 4), with dashed lines indicating linear regression fits. **Higher NTPS values correspond to lower MSE loss and higher accuracy in downstream tasks.**

### 4.1 CORRELATION BETWEEN NTPS AND LINEAR PROBE PERFORMANCE

First, we demonstrate that NTPS correlates with the downstream performance of eight pretrained models across 12 diverse datasets.

The experimental setup follows section 3.1, reusing the 12 datasets and the same set of models, but with two additional pretrained models (Qwen2-7B (Yang et al., 2024) and LLaMA3-8B (Grattafiori et al., 2024)).

Downstream performance is measured in two complementary ways. First, we train a linear layer on each downstream dataset using ordinary least-sqaure (OLS) regression with close-form solution since our theoretical derivation in theorem 1 is based on MSE loss (as shown in eq. (2) and eq. (3)). And we use the final MSE loss as the downstream performance metric. Second, we train a linear layer on each downstream dataset using logistic regression with saga optimizer under a CE loss since it better reflects practical usage. And we use the final accuracy as the downstream performance metric.

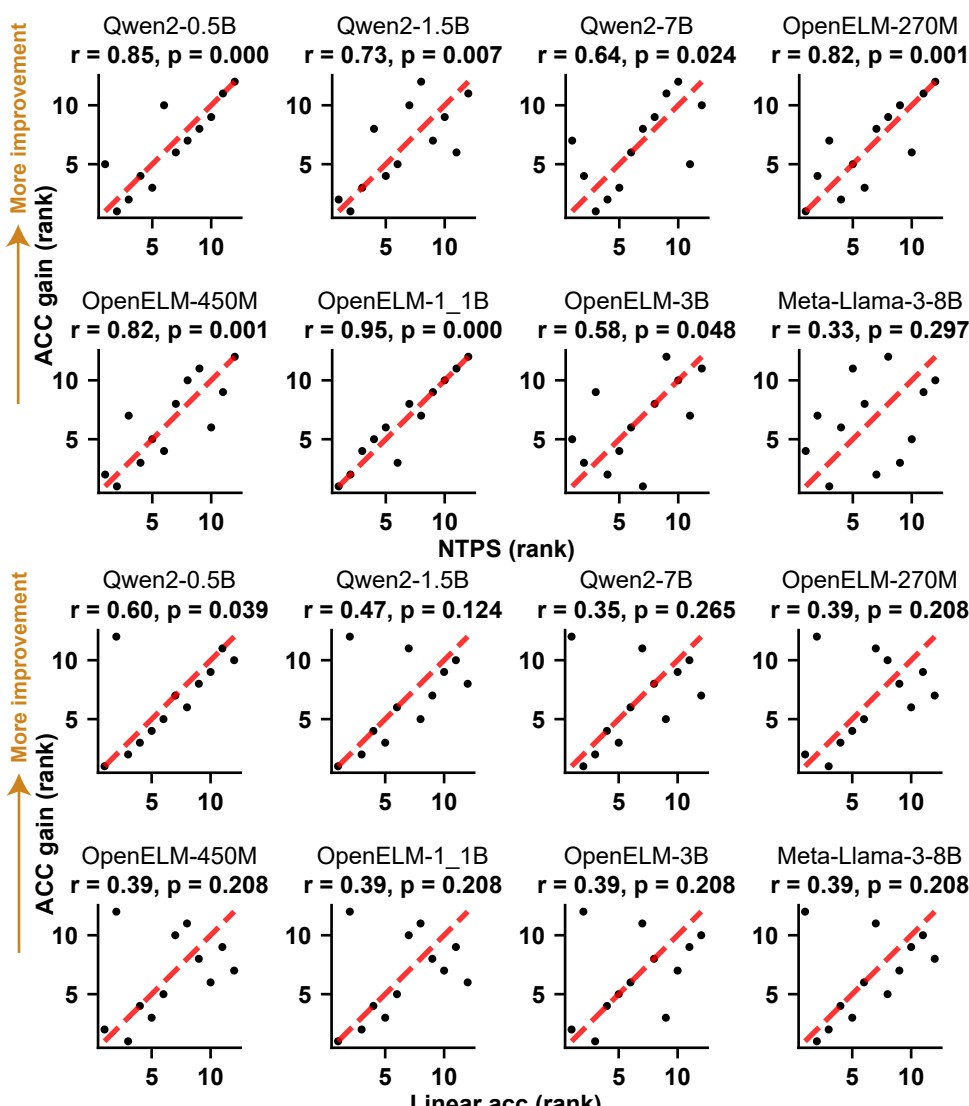

Figure 3: Correlation between NTPS vs. accuracy gain (LoRA finetuning accuracy-linear probing accuracy) and linear probing accuracy vs. accuracy gain, with dashed lines indicating linear regression fits. **NTPS is a better predictor of accuracy gain than linear probing accuracy with higher NTPS values correspond to lower accuracy gains after LoRA finetuning in downstream tasks.**

For each model across all datasets, we compute NTPS over all layers (from the word-embedding layer through the penultimate layer) and every $k$ proportion value from 0.05 to 0.95 in increments of 0.05. We then assess the monotonic relationship between NTPS and downstream performance using Spearman's $r \in [-1, 1]$. where $r = 1$ denotes perfectly concordant orderings.

To summarize each model succinctly, we report for each model the NTPS value corresponding to the configuration that yields the strongest Spearman's $r$. Besides, all results are obtained on training set to minimize confounding factors such as distribution shifts between train and test splits, which may obscure the true relationship between NTPS and downstream performance.

The results, shown in fig. 2, reveal a clear trend: higher NTPS values are associated with lower MSE losses and higher accuracies in downstream tasks. This strong correspondence indicates that NTPS, despite its derivation under simplified linear assumptions, serves as an effective proxy for task alignment even in highly nonlinear models. It provides insight into when autoregressive training is

beneficial for downstream tasks and can serve as a practical metric for anticipating the effectiveness of linear probing on pretrained models.

> **Takeaway**: NTPS shows a clear monotonic relationship with downstream linear probe performance in LLMs—higher NTPS predicts better performance.

### 4.2 PREDICTING LoRA FINETUNING GAIN WITH NTPS

Now, we are going to show that our NTPS can also serve for practical usage, particularly for predicting the LoRA finetuning gain. (See section A.6 for corroborating evidence that LoRA increases alignment: across eight models and 12 datasets, NTPS increases in 71/96 runs after LoRA).

To evaluate whether NTPS can forecast the benefit of parameter-efficient finetuning, we measure the "ACC gain" as the difference between accuracy after LoRA adaptation and the baseline linear-probe accuracy (both on the test split). We reuse the same eight pretrained models (Qwen2-0.5B/1.5B/7B, OpenELM-270M/450M/1.1B/3B, and LLaMA3-8B) and the 12 downstream classification tasks described in section 4.1. Again we (1) compute NTPS over all layers and $k$s exactly as before, (2) train a linear probe under CE to get baseline accuracy (AdamW optimizer, learning rate of $10^{-4}$; 50 epochs), and (3) apply LoRA (rank 32, $\alpha = 32$, 5000 steps, Adafactor, 5% warm-up) and record the adapted accuracy. Finally, we correlate NTPS with the observed LoRA gains using Spearman's $r$.

As plotted in fig. 3, there is a clear monotonic relationship: models with lower NTPS enjoy larger accuracy gains from LoRA, whereas higher NTPS exhibit only modest improvements. Across the eight models, Spearman's $r$ ranges from 0.40 up to 0.90 (higher absolute $r$ indicates stronger predictivity), confirming that NTPS is a reliable indicator of how much headroom remains for downstream adaptation. In comparison, linear probing accuracy only shows modest predictive power for accuracy gains from LoRA (Spearman's $r$ ranges from 0.30 up to 0.60), limiting its practical usage for LoRA finetuning prediction.

In practical terms, if a pretrained model yields a low NTPS on the target task, one can anticipate a sizable boost from LoRA; conversely, if a model yields a high NTPS, it is unlikely to benefit substantially from further finetuning. This makes NTPS a lightweight pre-screening tool to decide when parameter-efficient finetuning is most worthwhile.

> **Takeaway**: NTPS inversely predicts the accuracy gains from LoRA finetuning: tasks with low initial alignment see the largest boosts.

## 5 CONCLUSION

In conclusion, in this paper we have introduced *NTPS*, a simple yet powerful metric for measuring the alignment between the feature subspaces learned by autoregressive pretraining and those required for downstream perception tasks. In a linear setting, we proved that NTPS both upper- and lower-bounds the excess regression loss of an autoregressive encoder relative to an ideal perceptual encoder. Empirically, we demonstrate that NTPS, computed in closed form from pretrained representations and labeled data, correlates strongly with classification accuracy across 12 diverse NLP datasets and eight pretrained models ranging from 270 M to 8 B parameters. In addition, we examine a potential application of NTPS, predicting the accuracy gain after LoRA finetuning.

Our work still has several limitations that can be addressed in future research. First, NTPS is derived under a simplified linear setting. While sentence representations extracted from intermediate layers of nonlinear models allow our method to capture certain nonlinear effects, the theoretical formulation underlying theorem 1 remains linear. Extending the framework with kernel methods (e.g., the neural tangent kernel) could yield a fully nonlinear version and provide more precise characterizations. Second, we have not yet explored how to select optimal configurations for computing NTPS in each model beforehand. For example, the choice of $k$ may depend on the model's compression rate. Developing a more principled configuration strategy could improve efficiency and eliminate the need to exhaustively search over all possible settings.

# 6 REPRODUCIBILITY STATEMENT

We provide complete proofs for all lemmas and theorems (lemma 1 and theorems 1 and 2) in sections A.1 to A.3. The code for reproducing all experiments is included in the supplementary material.

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

# A  APPENDIX

## A.1  PROOF OF LEMMA 1

**Lemma 1** (Equivalence between MSE and CE; proof in section A.1, empirical validation in (Hui & Belkin, 2020)). *Let $X \in \mathbb{R}^{d \times \ell}$ be the token representations. Denote*

$$h^* = X L_2, \qquad \hat{h} = W^\top V^\top X L_1.$$

*Assume the vocabulary embeddings $\{w_i\}_{i=1}^V \subset \mathbb{R}^d$ satisfy a positive margin*

$$\Delta \;=\; \min_{j \neq y} \langle w_y, h^* \rangle - \langle w_j, h^* \rangle \;>\; 0, \tag{4}$$

*If $\mathcal{L} \;=\; \mathbb{E}\|\hat{h} - h^*\|_F^2 \;\to\; 0,$ then*

$$\Pr\!\big(\arg\max_i \langle w_i, \hat{h} \rangle = y\big) \;\longrightarrow\; 1. \tag{5}$$

*Proof.* For any $j \neq y$, write

$$\langle w_y, \hat{h} \rangle - \langle w_j, \hat{h} \rangle = \langle w_y, h^* \rangle - \langle w_j, h^* \rangle + \langle w_y - w_j, \hat{h} - h^* \rangle.$$

By definition the first term is at least $\Delta$, and by Cauchy–Schwarz

$$\big|\langle w_y - w_j, \hat{h} - h^* \rangle\big| \;\leq\; \|w_y - w_j\|_2 \|\hat{h} - h^*\|_2 \;\leq\; M \|\hat{h} - h^*\|_2,$$

where

$$M \;=\; \max_{i \neq j} \|w_i - w_j\|_2.$$

Hence

$$\langle w_y, \hat{h} \rangle - \langle w_j, \hat{h} \rangle \;\geq\; \Delta - M \|\hat{h} - h^*\|_2.$$

In particular, whenever $\|\hat{h} - h^*\|_2 < \Delta/M$ we have $\langle w_y, \hat{h} \rangle > \langle w_j, \hat{h} \rangle$ for all $j \neq y$, so the arg-max picks the correct token $y$. Since $\mathbb{E}\|\hat{h} - h^*\|_2^2 \to 0$ implies $\|\hat{h} - h^*\|_2 \to 0$ in probability, the probability of decoding error goes to zero. $\square$

## A.2  PROOF OF THEOREM 1

**Theorem 1** (proof in section A.2). *The loss functions $\mathcal{L}$ in eq. (2) and $\mathcal{L}^*$ in eq. (3) are minimized for*

$$W = (V^\top \mathbb{E}[X L_1 L_1^\top X] V)^{-1} V^\top \mathbb{E}[X L_1 L_2^\top X] \tag{6}$$

$$Z = (U^\top \mathbb{E}[X 1_{\ell \times 1} 1_{\ell \times 1}^\top X] U)^{-1} U^\top \mathbb{E}[X 1_{\ell \times 1} Y^\top] \tag{7}$$

*$U, V$ span the top $k$ eigenvectors of the following generalized eigenvalue problems:*

$$\mathbb{E}[X L_1 L_2^\top X^\top] \, \mathbb{E}[X L_2 L_1^\top X^\top] \, \tilde{V} = \mathbb{E}[X L_1 L_1^\top X^\top] \, \tilde{V} \, \Lambda_V, \tag{8}$$

$$\mathbb{E}[X 1_{\ell \times 1} Y^\top] \, \mathbb{E}[Y 1_{1 \times \ell} X^\top] \, \tilde{U} = \mathbb{E}[X 1_{\ell \times \ell} X^\top] \, \tilde{U} \, \Lambda_U. \tag{9}$$

*Proof.* Consider a sentence, whose token representation is $X \in \mathbb{R}^{(d, \ell)}$, where $d$ is the hidden dimension size of each token and $\ell$ is the total number of tokens in this sentence. Consider two variants $X_1, X_2 \in \mathbb{R}^{d \times (\ell - 1)}$ from $X$. For $X_1$, the $i$-th column represents the sum of the first $i$ tokens in $X$. For $X_2$, the $i$-th column represents the $(i+1)$-th token in $X$ so the whole matrix denotes the sequence of tokens from position 2 through $\ell$ in $X$.

Given a linear model to predict the $n$-th token given the sum of the previous $n-1$ tokens that contains a linear mapping $V \in \mathbb{R}^{d \times k}$ and a linear mapping $W \in \mathbb{R}^{k \times d}$. And set the loss $\mathcal{L}$ as:

$$\mathcal{L} = \mathbb{E}_{X_1, X_2}[\|W^\top V^\top X_1 - X_2\|_F^2], \tag{11}$$

Denote $A = V^\top X_1 \in \mathbb{R}^{k \times (\ell-1)}$, then:

$$\begin{aligned}
\mathcal{L} &= \mathbb{E}_{X_1,X_2}[\|W^\top A - X_2\|_F^2] \\
&= \mathbb{E}_{X_1,X_2}[Tr((W^\top A - X_2)^\top (W^\top A - X_2))] \\
&= \mathbb{E}_{X_1,X_2}[Tr(A^\top W W^\top A) - 2Tr(X_2^\top W^\top A) + Tr(X_2^\top X_2)]
\end{aligned} \tag{12}$$

Taking the derivative of $\mathcal{L}$ w.r.t. $W$.

$$\begin{aligned}
\frac{\partial \mathcal{L}}{\partial W} &= \frac{\partial \mathbb{E}_{X_1,X_2}[Tr(A^\top W W^\top A) - 2Tr(X_2^\top W^\top A) + Tr(X_2^\top X_2)]}{\partial W} \\
&= \frac{\partial \mathbb{E}_{X_1,X_2}[Tr(A^\top W W^\top A)]}{\partial W} - 2\frac{\partial \mathbb{E}_{X_1,X_2}[Tr(X_2^\top W^\top A)]}{\partial W} \\
&= \frac{\partial \mathbb{E}_{X_1,X_2}[Tr(W^\top A A^\top W)]}{\partial W} - 2\frac{\partial \mathbb{E}_{X_1,X_2}[Tr(W^\top A X_2^\top)]}{\partial W} \\
&= 2E_{X_1,X_2}[A A^\top W - A X_2^\top] \\
&= 2E_{X_1,X_2}[V^\top X_1 X_1^\top V W - V^\top X_1 X_2^\top] \\
&= 2E_{X_1,X_2}[V^\top X_1 (X_1^\top V W - X_2^\top)]
\end{aligned} \tag{13}$$

To minimize $\mathcal{L}$, we set $eq.\ (13) = 0$:

$$\begin{aligned}
2E_{X_1,X_2}[V^\top X_1 (X_1^\top V W - X_2^\top)] &= 0 \\
E_{X_1,X_2}[V^\top X_1 X_1^\top V W] &= E_{X_1,X_2}[V^\top X_1 X_2^\top]
\end{aligned} \tag{14}$$

Assume that $V^T X_1$ has rank $k$, then $V^T X_1 X_1^T V$ is invertible, and we can express $W$ from eq. (14) as below:

$$\begin{aligned}
W &= E_{X_1,X_2}[(V^\top X_1 X_1^\top V)^{-1}] E_{X_1,X_2}[V^\top X_1 X_2^\top] \\
W &= (V^\top \mathbb{E}[X_1 X_1^\top] V)^{-1} V^\top \mathbb{E}[X_1 X_2^\top]
\end{aligned} \tag{15}$$

By plugging in $W$ back to $\mathcal{L}$, we have:

$$\begin{aligned}
\mathcal{L} &= \mathbb{E}[Tr(X_1^T V (V^\top \mathbb{E}[X_1 X_1^\top] V)^{-1} V^\top \mathbb{E}[X_1 X_2^\top] \mathbb{E}[X_2 X_1^\top] V (V^T \mathbb{E}[X_1 X_1^T] V)^{-1} V^T X_1)] \\
&\quad - 2\mathbb{E}[Tr(X_2^\top \mathbb{E}[X_2 X_1^\top] V (V^\top \mathbb{E}[X_1 X_1^\top] V)^{-1} V^\top X_1)] + \mathbb{E}[Tr(X_2^\top X_2)] \\
&= Tr(\mathbb{E}[X_1^T V (V^\top \mathbb{E}[X_1 X_1^\top] V)^{-1} V^\top \mathbb{E}[X_1 X_2^\top] \mathbb{E}[X_2 X_1^\top] V (V^T \mathbb{E}[X_1 X_1^T] V)^{-1} V^T X_1]) \\
&\quad - 2Tr(\mathbb{E}[X_2^\top \mathbb{E}[X_2 X_1^\top] V (V^\top \mathbb{E}[X_1 X_1^\top] V)^{-1} V^\top X_1]) + Tr(\mathbb{E}[X_2^\top X_2]) \\
&= Tr(\mathbb{E}[V^T X_1 X_1^T V (V^\top \mathbb{E}[X_1 X_1^\top] V)^{-1} V^\top \mathbb{E}[X_1 X_2^\top] \mathbb{E}[X_2 X_1^\top] V (V^T \mathbb{E}[X_1 X_1^T] V)^{-1}]) \\
&\quad - 2Tr(\mathbb{E}[\mathbb{E}[X_2 X_1^\top] V (V^\top \mathbb{E}[X_1 X_1^\top] V)^{-1} V^\top X_1 X_2^\top]) + Tr(\mathbb{E}[X_2^\top X_2]) \\
&= Tr((V^\top \mathbb{E}[X_1 X_1^\top] V)(V^\top \mathbb{E}[X_1 X_1^\top] V)^{-1} V^\top \mathbb{E}[X_1 X_2^\top] \mathbb{E}[X_2 X_1^\top] V (V^\top \mathbb{E}[X_1 X_1^\top] V)^{-1}) \\
&\quad - 2Tr(\mathbb{E}[X_2 X_1^\top] V (V^\top \mathbb{E}[X_1 X_1^\top] V)^{-1} V^\top \mathbb{E}[X_1 X_2^\top]) + Tr(\mathbb{E}[X_2^\top X_2]) \\
&= Tr(\mathbb{E}[X_2 X_1^\top] V (V^\top \mathbb{E}[X_1 X_1^\top] V)^{-1} V^\top \mathbb{E}[X_1 X_2^\top]) \\
&\quad - 2Tr(\mathbb{E}[X_2 X_1^\top] V (V^\top \mathbb{E}[X_1 X_1^\top] V)^{-1} V^\top \mathbb{E}[X_1 X_2^\top]) + Tr(\mathbb{E}[X_2^\top X_2]) \\
&= Tr(\mathbb{E}[X_2^\top X_2]) - Tr(\mathbb{E}[X_2 X_1^\top] V (V^\top \mathbb{E}[X_1 X_1^\top] V)^{-1} V^\top \mathbb{E}[X_1 X_2^\top])
\end{aligned} \tag{16}$$

Denote $R_{11} = \mathbb{E}[X_1 X_1^\top]$, $R_{12} = \mathbb{E}[X_1 X_2^\top]$, $R_{22} = \mathbb{E}[X_2 X_2^\top]$, minimizing $\mathcal{L}$ is solving the following maximization problem:

$$\max_V Tr(V^\top R_{12} R_{12}^\top V (V^\top R_{11} V)^{-1}) \tag{17}$$

which is equivalent to the following maximization problem:

$$\max_{\tilde{V}:\tilde{V}^\top R_{11} \tilde{V} = I} Tr(\tilde{V}^\top R_{12} R_{12}^\top \tilde{V}) \tag{18}$$

And we can observe that the constraint is satisfied when:

$$\tilde{V} = V(V^\top R_{11} V)^{-\frac{1}{2}} \tag{19}$$

Thus, $\tilde{V}$ and $V$ share the same column space. And the subspace can be found via the optimization problem in eq. (18), which yields to the generalized eigenvalue problem Ghojogh et al. (2019):

$$R_{12} R_{12}^\top \tilde{V} = R_{11} \tilde{V} \Lambda \tag{20}$$

Since eq. (18) is a maximization problem, $\tilde{V}$ contains the eigenvectors of $(R_{12} R_{12}^\top, R_{11})$ that correspond to the top $k$ largest eigenvalues. And so $V$ spans the same column space as these eigenvectors.

From our definition we have $X_1 = XL_1, X_2 = XL_2$ with $L_1, L_2 \in \mathbb{R}^{(\ell, \ell-1)}$ defined as:

$$L_1 = \begin{bmatrix} Q_{\ell-1} \\ 0 \end{bmatrix}, \tag{21}$$

$$L_2 = \begin{bmatrix} 0 \\ I_{\ell-1} \end{bmatrix}, \tag{22}$$

where $Q_{l-1} \in \mathbb{R}^{l-1 \times l-1}$ is a unit upper triangular matrix (i.e. all entries on or above the diagonal are 1 and 0 below the diagonal).

Denote

$$D \in \mathbb{R}^{\ell \times \ell} = L_1 L_1^\top = \begin{bmatrix} Q_{\ell-1} Q_{\ell-1}^\top & 0 \\ 0 & 0 \end{bmatrix} \tag{23}$$

$$S \in \mathbb{R}^{\ell \times \ell} = L_1 L_2^\top = \begin{bmatrix} 0 & Q_{\ell-1} \\ 0 & 0 \end{bmatrix} \tag{24}$$

Then we can rewrite the generalized eigenvalue problem in eq. (20) as:

$$\mathbb{E}[XSX^\top]\mathbb{E}[XS^\top X^\top]\tilde{V} = \mathbb{E}[XDX^\top]\tilde{V}\Lambda \tag{25}$$

Now let's consider a regression task with the label of the sentence $X$ denoted as $Y \in \mathbb{R}^c$.

Given a linear model to predict the label based on the sum of all tokens in the sentence that contains a linear mapping $U \in \mathbb{R}^{d \times k}$ and $Z \in \mathbb{R}^{k \times c}$. And set the learning objective as mean squared error (MSE) loss $\mathcal{L}^*$ as defined below:

$$L^* = \mathbb{E}\left\| Z^\top U^\top X \mathbf{1}_{\ell \times 1} - Y \right\|_F^2, \tag{26}$$

where $\mathbf{1}_{\ell \times 1} \in \mathbb{R}^{\ell \times 1}$ is for summing the tokens in $X$.

Similarly, with the optimal $Z$, we will have the optimal $U$ sharing the same column space as $\tilde{U}$ that contains the eigenvectors corresponding to the largest k eigenvalues of the following generalized eigenvalue problem:

$$\mathbb{E}[X\mathbf{1}_{\ell \times 1}Y^\top]\mathbb{E}[Y(X\mathbf{1}_{\ell \times 1})^\top]\tilde{U} = \mathbb{E}[(X\mathbf{1}_{\ell \times 1})(X\mathbf{1}_{\ell \times 1})^\top]\tilde{U}\Lambda, \tag{27}$$

which simplifies to:

$$\mathbb{E}[X\mathbf{1}_{\ell \times 1}Y^\top]\mathbb{E}[Y\mathbf{1}_{1 \times \ell}X^\top]\tilde{U} = \mathbb{E}[X\mathbf{1}_{\ell \times \ell}X^\top]\tilde{U}\Lambda, \tag{28}$$

$\square$

### A.3  Proof of Theorem 2

**Theorem 2** (Excess regression loss bounded by NTPS; proof in section A.3). *Let $U$ be the optimal perceptual encoder obtained from the generalized eigenproblem. For any other encoder $V$, define the orthogonal projector. Let $\Delta\mathcal{L} := \mathcal{L}^*(V) - \mathcal{L}^*(U)$ be the excess regression loss of $V$.*

*Then there exist task-dependent positive constants $C_{\min}, C_{\max}$ such that*

$$\boxed{C_{\min}\big(1 - \mathrm{NTPS}(U,V)\big) \ \leq \ \Delta\mathcal{L} \ \leq \ C_{\max}\big(1 - \mathrm{NTPS}(U,V)\big).} \tag{10}$$

*Proof.* Denote $N := \mathbb{E}[X 1_{\ell\times\ell} X^\top]$, $M := \mathbb{E}[X 1_{\ell\times 1} Y^\top]$.

$$\mathcal{L}^*(V) = \mathrm{Tr}(\mathbb{E}[YY^\top]) - \mathrm{Tr}\big(V^\top MM^\top V \, (V^\top NV)^{-1}\big). \tag{29}$$

$$\mathcal{L}^*(U) = \mathrm{Tr}(\mathbb{E}[YY^\top]) - \mathrm{Tr}\big(U^\top MM^\top U \, (U^\top NU)^{-1}\big). \tag{30}$$

Recall that the columns of $U$ solve the generalized eigenvalue problem, and $U$ is $N$–orthonormal $\big(U^\top NU = I_k\big)$

$$MM^\top U \ = \ NU\Lambda, \qquad \Lambda = \mathrm{diag}(\Lambda_{11}, \dots, \Lambda_{kk}), \ \Lambda_{11} \geq \cdots \geq \Lambda_{kk} > 0. \tag{31}$$

the minimal regression loss is

$$\mathcal{L}^*(U) = \mathrm{Tr}(\mathbb{E}[YY^\top]) - \sum_{i=1}^{k} \Lambda_{ii}. \tag{32}$$

Premultiplying eq. (31) by $N^{-1/2}$ and defining the whitened basis $U^* := N^{1/2}U$ gives the *ordinary symmetric eigenproblem*. ($U$ and $U^*$ share the same subspace)

$$N^{-1/2} MM^\top N^{-1/2} U^* \ = \ U^* \Lambda. \tag{33}$$

Introduce the whitened encoder $V^* := N^{1/2}V \, (V^\top NV)^{-1/2}$

$$\begin{aligned}
T(V) :&= \mathrm{Tr}\big(V^\top MM^\top V \, (V^\top NV)^{-1}\big) \\
&= \mathrm{Tr}\big(V^{*\top} N^{-1/2} MM^\top N^{-1/2} V^*\big) \\
&= \mathrm{Tr}\big(V^{*\top} U^* \Lambda U^{*\top} V^*\big) \\
&= \sum_{i=1}^{k} \Lambda_{ii} \, \|V^{*\top} u_i^*\|_2^2, \\
&= \sum_{i=1}^{k} \Lambda_{ii} \, u_i^{*\top} V^* V^{*\top} u_i^*, \\
&= \sum_{i=1}^{k} \Lambda_{ii} \, u_i^\top N^{1/2\top} V^* V^{*\top} N^{1/2} u_i, \\
&= \sum_{i=1}^{k} \Lambda_{ii} \, u_i^\top NV \, (V^\top NV)^{-1} V^\top N \, u_i
\end{aligned} \tag{34}$$

$$\begin{aligned}
\Delta\mathcal{L} :&= \sum_{i=1}^{k} \Lambda_{ii} \, \big(1 - \|V^{*\top} u_i^*\|_2^2\big) \\
&= \sum_{i=1}^{k} \Lambda_{ii} \, \big(1 - u_i^{*\top} V^* V^{*\top} u_i^*\big) \\
&= \sum_{i=1}^{k} \Lambda_{ii} \, \big(1 - u_i^\top NV \, (V^\top NV)^{-1} V^\top N \, u_i\big).
\end{aligned} \tag{35}$$

Note that

$$
\begin{aligned}
\text{NTPS} &= \frac{\|PU\|_F^2}{\|U\|_F^2} \\
&= \frac{\operatorname{Tr}\big(U^\top P^\top P U\big)}{\|U\|_F^2} \\
&= \frac{\operatorname{Tr}\big(U^\top P U\big)}{\|U\|_F^2} \\
&= \frac{\sum_{i=1}^k u_i^\top P u_i}{\|U\|_F^2} \\
&= \frac{\sum_{i=1}^k u_i^\top V \, (V^\top V)^{-1} V^\top \, u_i}{\|U\|_F^2}.
\end{aligned}
\tag{36}
$$

For each $i$, set

$$
r_i = 1 - u_i^\top N V (V^\top N V)^{-1} V^\top N u_i.
$$

Writing $w = Va$ and minimizing

$$
(u_i - Va)^\top N (u_i - Va) = u_i^\top N u_i - 2a^\top V^\top N u_i + a^\top V^\top N V a
$$

over $a$ yields $a^* = (V^\top N V)^{-1} V^\top N u_i$, so

$$
r_i = u_i^\top N u_i - u_i^\top N V (V^\top N V)^{-1} V^\top N u_i = \min_{w \in \operatorname{col}(V)} (u_i - w)^\top N (u_i - w).
$$

Since $x^\top N x \geq \lambda_{\min}(N)\|x\|^2$,

$$
\begin{aligned}
r_i &\geq \lambda_{\min}(N) \min_{w \in \operatorname{col}(V)} \|u_i - w\|^2, \\
&= \lambda_{\min}(N)[\|u_i\|^2 - u_i^\top V (V^\top V)^{-1} V^\top u_i].
\end{aligned}
\tag{37}
$$

Thus

$$
\begin{aligned}
\Delta\mathcal{L} &\geq \lambda_{\min}(N) \sum_i \Lambda_{ii}[\|u_i\|^2 - u_i^\top V (V^\top V)^{-1} V^\top u_i], \\
&\geq \lambda_{\min}(N)\Lambda_{\min}\|U\|_F^2(1 - \text{NTPS}),
\end{aligned}
\tag{38}
$$

Similarly, $x^\top N x \leq \lambda_{\max}(N)\|x\|^2$ gives

$$
r_i \leq \lambda_{\max}(N)[\|u_i\|^2 - u_i^\top V (V^\top V)^{-1} V^\top u_i],
\tag{39}
$$

and hence

$$
\Delta\mathcal{L} \leq \lambda_{\max}(N)\Lambda_{\max}\|U\|_F^2(1 - \text{NTPS}).
\tag{40}
$$

Combining,

$$
\boxed{\lambda_{\min}(N)\Lambda_{\min}\|U\|_F^2\,(1 - \text{NTPS}) \leq \Delta\mathcal{L} \leq \lambda_{\max}(N)\Lambda_{\max}\|U\|_F^2\,(1 - \text{NTPS})}
\tag{41}
$$

$\square$

## A.4 PSEUDOCODE FOR NTPS

---

**Algorithm 1** Computation of NTPS

---

**Require:** Dataset $\mathcal{D} = \{(x_i, y_i)\}_{i=1}^n$, pretrained transformer $f$, tokenizer $\mathcal{T}$, hidden dimension $d$, subspace dimension $k$, target layer $l$

**Ensure:** NTPS at layer $l$ using top-$k$ subspace

1: Initialize: `meanXX` $\in \mathbb{R}^{d \times d}$, `meanXY` $\in \mathbb{R}^{d \times c}$, `cov0` $\in \mathbb{R}^{d \times d}$, `cov1` $\in \mathbb{R}^{d \times d}$
2: **for** each sample $(x, y) \in \mathcal{D}$ **do**
3:      Tokenize $x$ with $\mathcal{T}$ to obtain input IDs and attention mask
4:      Run forward pass of $f$ to obtain token-level hidden states $X^l \in \mathbb{R}^{\ell \times d}$ at layer $l$
5:      Construct $L_1 \in \mathbb{R}^{\ell \times (\ell-1)}$ (upper-triangular), $L_2 \in \mathbb{R}^{\ell \times (\ell-1)}$ (lower-shifted identity)
6:      One-hot encode label $y \to Y \in \mathbb{R}^c$
7:      Compute mean token representation: $\bar{X}^l = \frac{1}{\ell} \sum_{j=1}^\ell X_j^l \in \mathbb{R}^d$
8:      `meanXX` $+= \frac{1}{n} \bar{X}^l (\bar{X}^l)^\top$
9:      `meanXY` $+= \frac{1}{n} \bar{X}^l Y^\top$
10:     `cov0` $+= \frac{1}{n} (X^l)^\top L_1 L_1^\top X^l$
11:     `cov1` $+= \frac{1}{n} (X^l)^\top L_1 L_2^\top X^l$
12: **end for**
13: Compute top generalized eigenspace $U$ from (`meanXY meanXY`$^\top$, `meanXX`)
14: Compute top generalized eigenspace $V$ from (`cov1 cov1`$^\top$, `cov0`)
15: Extract top-$k$ directions: $U_k \leftarrow$ first $k$ columns of $U$, $V_k \leftarrow$ first $k$ columns of $V$
16: Compute projection: $P_k \leftarrow V_k (V_k)^+$
17: Compute NTPS: `NTPS` $\leftarrow \|P_k U_k\|_F^2 / \|U_k\|_F^2$
18: **return** `NTPS`

---

## A.5 TRAINING LOSS

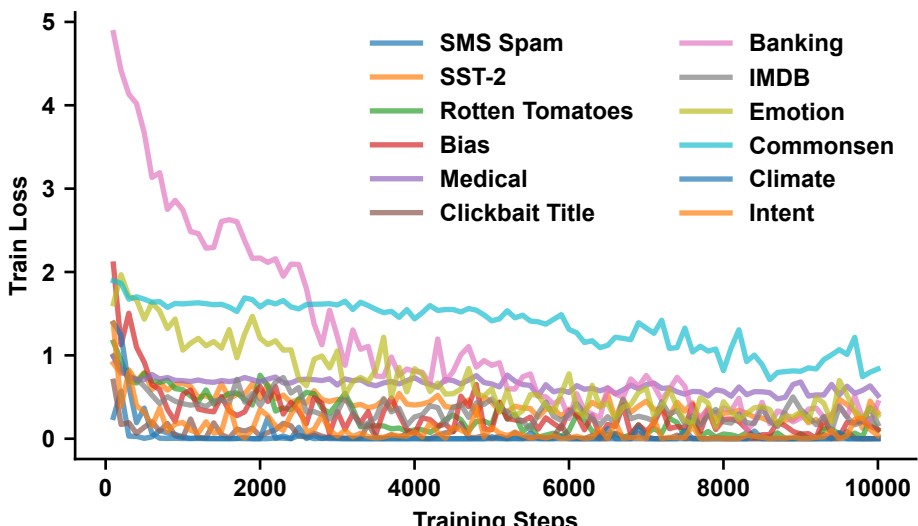

Figure S1: Cross entropy loss across training steps across datasets average across models. **The majority converges around 7K-8K steps, and therefore 10K steps provide sufficient budget to converge.**

## A.6 LoRA FINETUNING ENHANCES NTPS

As a side note, we provide an interpretation of why LoRA is effective for adapting pretrained LLMs to downstream tasks, through the lens of NTPS.

Table 2: Relative improvement (%) of NTPS after LoRA finetuning across models and downstream datasets. **NTPS is universally increased after LoRA finetuning, suggesting that LoRA finetuning enhances the overlap between feature subspaces of autoregressive training and downstream perception tasks.** For small models, NTPS slightly decreases probably because these models sacrifice the next-token prediction for higher downstream performance due to their limited capability.

| Dataset | Qwen2 0.5B | Qwen2 1.5B | Qwen2 7B | OpenELM 270M | OpenELM 450M | OpenELM 1.1B | OpenELM 3B | LlaMA-3 8B |
|---|---|---|---|---|---|---|---|---|
| Intent | 1.9 | 1.6 | 53.2 | -1.1 | -1.0 | 0.9 | 88.5 | 77.8 |
| Clickbait Title | 1.7 | 1.9 | 70.1 | -0.6 | -0.3 | -0.5 | 74.8 | 74.3 |
| SST-2 | 1.4 | 4.3 | 105.9 | -0.8 | -0.7 | 2.1 | 100.0 | 102.5 |
| Banking | 2.3 | 0.9 | 67.8 | -0.8 | -0.6 | 0.7 | 79.5 | 87.1 |
| Bias | 3.1 | 3.6 | 78.3 | -1.4 | -1.4 | -0.4 | 73.3 | 81.3 |
| Emotion | 2.3 | 2.9 | 109.0 | -1.5 | -2.3 | 0.5 | 83.4 | 80.0 |
| SMS Spam | 0.7 | -0.1 | 124.1 | -1.4 | -0.7 | 0.1 | 78.2 | 79.0 |
| Medical | 2.2 | 2.3 | 120.5 | -0.8 | -0.2 | 0.7 | 228.7 | 92.9 |
| Rotten Tomatoes | 0.8 | 0.2 | 105.6 | -1.6 | -0.4 | 0.1 | 88.6 | 84.3 |
| Commonsense | 1.1 | 0.7 | 108.8 | 1.1 | 0.6 | 1.0 | 91.6 | 95.0 |
| Climate | 1.5 | 1.8 | -14.5 | -0.4 | -1.8 | 1.2 | 76.7 | 99.9 |
| IMDB | 0.4 | -0.5 | 135.1 | 0.9 | 1.5 | 1.0 | 109.1 | 89.4 |

Specifically, we compute NTPS for the same eight models and 12 datasets used in section 4.1, using the exact same configuration for NTPS computation, but after applying LoRA. We adopt a consistent finetuning setup across all experiments: LoRA is applied to all QKV projection layers with rank 32, $\alpha = 32$, no bias, and a dropout rate of 0.05. For each input, we extract the mean token representation from the final transformer block and pass it into a linear classification head. We use the Adafactor optimizer with a learning rate of $10^{-4}$, $\epsilon$ values of $10^{-30}$ and $10^{-3}$, gradient clipping threshold of 1.0, decay rate of 0.8, and weight decay of $10^{-5}$. Training is conducted for 5000 steps with a cosine learning rate scheduler and a 5% warm-up phase.

As shown in table 2, NTPS increases in *71 out of 96* runs after applying LoRA. This provides empirical support for our interpretation: LoRA may improve downstream task performance by adjusting the representations to better align the feature subspaces used for autoregressive pretraining and those required for downstream tasks, especially in large models. Interestingly, we do see that NTPS slightly decreases in small models like OpenELM-270M and OpenELM-450M, this is probably because these model sacrifice the next-token prediction capability in exchange for higher downstream performance due to its limited capability.

### A.7 LLM USAGE DISCLOSURE

Large Language Models (LLMs) were used to assist with improving the clarity and readability of the manuscript. Specifically, LLM-based tools were employed for light language polishing, such as refining grammar and enhancing phrasing, without altering the underlying content or meaning. The core ideas, analysis, and writing were developed by the authors.

