# OpenReview forum: "Next Token Perception Score: Analytical Assessment of your LLM Perception Skills"
_ICLR.cc/2026/Conference — Submitted to ICLR 2026_

### Official Review · Reviewer_9jju · 2025-10-31

**Soundness:** 2
**Presentation:** 2
**Contribution:** 2
**Rating:** 4
**Confidence:** 3

**Summary:**

The paper introduces the Next-Token Perception Score (NTPS) to quantify how well features learned by next-token pretraining align with supervised “perception” tasks. NTPS measures the overlap between representations optimized for next-token prediction and those needed for labels, is computable in closed form from pretrained representations plus labeled data, and is proven to bound the excess loss (in a linear setting).  Empirically, the paper shows that linear-probe accuracy on pretrained models can outperform, match, or underperform full training from scratch across diverse datasets, motivating the need for such an alignment measure. The paper then shows that, across eight pretrained models and twelve datasets, NTPS monotonically predicts linear probing performance and predicts the accuracy gains from LoRA finetuning.

**Strengths:**

- A new metric, NTPS, to evaluate the misalignment between the pre-trained LM's feature and the features that are required for the downstream task.
- NTSP has theoretical guarantees (upper and lower bounds) on the excess loss for the MSE.
- Empirical results show that NTPS correlates well with the linear probe performance

**Weaknesses:**

- **Uncommon and narrow fine-tuning methods**: The paper extracts the mean last layer output and uses the mean embedding feed into a linear classifier head. This is significantly different from how we fine-tune LLMs nowadays: using prompting to make the model output natural language. It is unclear how the results may transfer to realistic fine-tuning settings that are widely used. A natural question is to ask whether the more natural fine-tuning setting (or older fine-tuning methods like pattern exploitation training) can make the autoregressive features become more aligned with the perception task's space.
- **Underspecified theoretical results**: The constants in Theorem 2 are not well discussed. If the constant is very large, the inequality may be less meaningful. While the empirical results in Section 4.1 may try to validate the theoretical results in the previous section, some assumptions (e.g., linear encoder) do not match, so Section 4 is more like empirical results instead of a justification of the theories.
- **Only reports the rank correlation**: The paper only reports the rank correlation between NTPS, accuracy, and MSE. For practical usages, a user may be more interested in how much absolute accuracy gain they can obtain, instead of whether the accuracy gain can be more than another task fine-tuned using the same model.
- **Unclear motivation of using LoRA fine-tuning**: It is unclear why LoRA fine-tuning is introduced in section 4.2. The main theory part of the paper discusses linear mapping, while Section 4.2 moves to a non-linear setting. I assume that the paper is trying to introduce a more natural setting, but the fine-tuning still uses the mean representation (the paper actually did not precisely describe the setting here, so I assume it is the same as the previous). This is still not a natural setting. Another question I have is why we are interested in the performance of LoRA instead of full fine-tuning. I would like to see some justification for why LoRA is selected here.

**Questions:**

- Q1. It seems that the term "perception task" is a term introduced in this paper, and not very common in NLP. This creates a severe understanding issue.

---

### Official Review · Reviewer_HnJc · 2025-10-31

**Soundness:** 3
**Presentation:** 3
**Contribution:** 3
**Rating:** 4
**Confidence:** 3

**Summary:**

The paper introduces the Next Token Perception Score (NTPS) that measures the overlap between autoregressive and perception feature subspaces, motivated by the idea that features from the autoregressive subspace may not be useful for perception tasks. The authors show that NTPS correlates strongly with a linear probe and predicts whether a LoRA finetuning can provide additional accuracy gains.

**Strengths:**

The paper is nicely organized and clearly motivated. The proposed NTPS metric provides an intuitive geometric perspective on alignment and the theoretical section is clear and builds good intuition. The experiments cover a wide range of models and datasets and NTPS shows convincing correlation with MSE loss and additional accuracy gains from LoRA.

**Weaknesses:**

All reported results are based on rank correlations, which makes it hard to interpret what the metric actually means in practice. If I have a model and a downstream task and compute an NTPS score, how should I interpret its magnitude? The paper doesn’t provide guidance on what constitutes a "high" or "low" score, which limits its usefulness.
The claim on syntactic vs semantic groupings (as in Fig 1) is nice for intuition but does not seem rigorous based on comparing just the top 2 eigenvalues.

**Questions:**

My main question is on how to use NTPS in practice -- given an individual downstream task how would I interpret the NTPS score?
Also, it would be helpful to include some discussion or analysis of which k values and which layers tend to perform best across different models and tasks. Are there consistent trends -- for example, are middle layers generally more aligned, or does the optimal k scale with model size?
The correlations are computed after sweeping over both k and layer, and then selecting the configuration that maximizes correlation on the training set. Do results hold if you test the resulting correlation on a held-out validation set?

---

### Official Review · Reviewer_KRVc · 2025-10-31

**Soundness:** 2
**Presentation:** 3
**Contribution:** 2
**Rating:** 4
**Confidence:** 3

**Summary:**

This paper investigates the misalignment between autoregressive pretraining and downstream perception tasks in large language models. The authors demonstrate that linear probe performance varies substantially across tasks, suggesting that features optimized for next-token prediction often do not transfer well to perception objectives. To address this, they introduce the Next Token Perception Score (NTPS), a metric measures the overlap between autoregressive feature subspace (V) and perception feature subspace (U). Theoretically, they prove that NTPS both upper- and lower-bounds the excess regression loss (Theorem 2), establishing it as a valid proxy for downstream performance. Empirically, the authors validate NTPS across 12 diverse NLP datasets and 8 pretrained models (270M to 8B parameters), demonstrating strong correlations with linear probe accuracy and MSE loss. Finally, they show that NTPS reliably predicts accuracy gains from LoRA finetuning.

**Strengths:**

1. The work proposes NTPS as a novel metric for measuring the misalignment between perception and next-token prediction objectives, addressing an important gap in understanding of pretrained LLMs’ limited transferability to downstream tasks.

2. The paper includes comprehensive and extensive experimental results : (1) Table 1 demonstrates that linear probing can outperform, match, or underperform full training from scratch, establishing the motivation for the work (2) Figure 2 shows consistent correlations between NTPS values and both MSE loss and accuracy across 8 models (3) Figure 3 demonstrates NTPS's utility in predicting LoRA finetuning gains (4) Experiments span diverse model families (Qwen2, LLaMA-3, OpenELM) and model scales (270M-8B parameters)

**Weaknesses:**

1. The paper claims that misalignment between perception and autoregressive spaces arises primarily from the next-token prediction loss during pretraining (lines 54-59, Section 3.1). However, other confounding factors could contribute to this phenomenon, including (1) pretraining data size and distribution mismatches with downstream tasks (2) optimization dynamics and implicit biases.

2. Related to the first point, the paper does not adequately control for or discuss these alternative explanations, weakening the causal claims about the source of misalignment.

3. The paper has many missing experimental details and lack of methodological clarity. In particular, the paper lacks how U is computed in practice: While Algorithm 1 in the appendix describes NTPS computation, the main text (Sections 3.2, 4.1) does not clearly specify which layer representations are used to compute U, how the subspace dimension k is selected, or how these choices affect results.

4. The narrative and flow are weak and often incoherent. Section 3.2 suggests some empirical evidence for the strength of NTPS, but the justification is not theoretically grounded or conceptually insightful. The experiment demonstrating NTPS as a proxy for LoRA fine-tuning (Section 4.2) is also not well motivated. The paper would benefit from clearer signposting and more explicit connections between theoretical claims and empirical validation.

**Questions:**

1. Conceptual justification for "perception": In the introduction of NTPS in Section 3.2, what is the theoretical or empirical motivation for terming U the "perception" subspace? The formulation in Eq. 9 appears to be a standard supervised dimensionality reduction via generalized eigen decomposition. What distinguishes this as specifically capturing "perception" as opposed to other notions?

2. Alternative explanations for misalignment: The paper attributes misalignment primarily to the next-token prediction objective during pretraining. However, could other factors explain the variability in Table 1, such as: training data distribution or optimization dynamics? How do the authors rule out these alternative explanations?

3. Acquisition of U in experiments: The main body of the text lacks clear specification of how U is acquired in each experiment. Specifically which layer's representations are used to compute U? And How is the subspace dimension k chosen (fixed value, cross-validation, proportion of variance)?

4. Motivation for LoRA finetuning analysis (Section 4.2): Why focus specifically on LoRA for demonstrating NTPS's practical utility?

---

### Official Review · Reviewer_nYiu · 2025-11-01

**Soundness:** 3
**Presentation:** 3
**Contribution:** 2
**Rating:** 6
**Confidence:** 3

**Summary:**

The authors propose a new measure NTPS as a predictive measure for downstream performance on different tasks. It quantifies the overlap between the feature subspace favored by next-token prediction and the subspace optimal for the downstream task. They show that this correlates with the actual downstream performances much better than the linear probe, and is a computationally efficient way of gauging the "fit" of the LLM for the given downstream task.

**Strengths:**

- The proposed metric and its derivation seem pretty clear. It is essentially a subspace alignment score between the frobenius norm of the perception encoder U that lies inside the next token subspace spanned by V.
- The metric seems to be well-correlated with downstream performance across different models.

**Weaknesses:**

> Takeaway: Linear probing on pretrained LLM representations can outperform, match, or underperform full-training from scratch.
- Agreed that the linear probing technique is indeed noisy, but Table 1 and this claim seem to be slightly misleading. These linear probes are used as a way to approximate how good the model are at the downstream tasks like Emotion, etc, so a better study seems to be how well the linear probes correlate to the full finetuning performance (when taking different checkpoints for example), not if they are better, worse, or equal.
- For Figures 2 and 3, can we also plot how the linear probe performance correlates with the MSE performance?

**Questions:**

- Are the results in Figure 2 and Figure 3 reported over all the downstream tasks in Table 1? What does each point represent?

---

### Meta-Review · Area_Chair_bWR7 · 2026-01-05

**Summary:**

The submission proposes Next Token Perception Score (NTPS), a closed-form subspace-overlap metric intended to quantify how well an autoregressive next-token subspace aligns with a downstream “perception” subspace, with a linear-theory guarantee (NTPS bounds excess loss) and empirical correlations across multiple models/datasets. Given the absence of a rebuttal, these issues remain unresolved, leading to an overall recommendation of rejection.

**Reviewer Concerns:**

Underspecified methodology: The paper lacks clarity on how the perception subspace  $U$ is computed (layer choice, subspace dimension $k$, sensitivity), and correlations are reported after sweeping over configurations, raising reproducibility and selection-bias concerns.

Limited practical interpretability: Results rely almost entirely on rank correlations, with little guidance on how to interpret absolute NTPS values or use them to predict concrete downstream gains.

Overstated causal framing: Misalignment is primarily attributed to next-token pretraining without sufficiently controlling for alternative factors such as data distribution, model scale, or optimization effects.

Questionable external validity: The “perception task” formulation and linear-head evaluation differ from common prompting or generation-based fine-tuning, making real-world applicability unclear.

**Reviewer Scores:**

nYiu: likely stays 6
KRVc: likely stays 4
HnJc: likely stays 4
9jju: likely stays 4

---

### Decision · Program_Chairs · 2026-01-26

Reject